# Erythroblasts Promote the Development of a Suppressive Lymphocyte Phenotype via Treg Induction and PD1 Upregulation on the Surfaces of B-Cells: A Study on the Subpopulation-Specific Features of Erythroblasts

**DOI:** 10.3390/cimb47070550

**Published:** 2025-07-15

**Authors:** Kirill Nazarov, Roman Perik-Zavodskii, Julia Shevchenko, Sergey Sennikov

**Affiliations:** Laboratory of Molecular Immunology, Federal State Budgetary Scientific Institution “Research Institute of Fundamental and Clinical Immunology”, 630099 Novosibirsk, Russia; kirill.lacrimator@mail.ru (K.N.); shevchenkoja2023@yandex.ru (J.S.)

**Keywords:** erythroid cells, erythroblasts, Treg, immunosuppression

## Abstract

This study identifies the novel effects of soluble factors derived from murine erythroblasts on lymphoid cell phenotypes. These effects were observed following the treatment of splenic mononuclear cells with erythroblast-conditioned media received from both healthy mice and mice subjected to hematopoiesis-activating conditions (hypoxia, blood loss, and hemolytic anemia), suggesting a common mechanism of action. Using flow cytometry, we elucidated that erythroblast-derived soluble products modulate T cell differentiation by promoting Treg development and increasing PD-1 surface expression on B cells. The immunoregulatory potential of erythroblasts is subpopulation-dependent: CD45+ erythroblasts respond to hemolytic stress by upregulating the surface expression of immunosuppressive molecules PDL1 and Galectin-9, while CD45- erythroblasts primarily increase TGFb production. These findings highlight the regulatory role of erythroblasts in modulating immune responses.

## 1. Introduction

It is well-established that conditions activating erythropoiesis are frequently accompanied by immunosuppression. In mice exposed to acute hypoxia before and after immunization with sheep erythrocytes, an increased proliferation of erythroid progenitors in the spleen was observed along with a delay in the development of the humoral immune response [1]. Early studies also demonstrated the ability of erythroblasts to inhibit B cell proliferation and suppress the humoral immune response [2,3]. However, the molecular mechanisms of the influence of erythroblasts on lymphoid cells remain poorly understood.

Phenylhydrazine-induced hemolytic anemia in mice results in enhanced extramedullary erythropoiesis, marked by increased splenic BFU-E colonies [4]. Hemolytic anemia triggers an expansion of stress BFU-E populations in the spleen while reducing BFU-E generation in the bone marrow—a common feature of various experimental anemia models, including those induced by sterile inflammation [5].

CD45+ erythroblasts have been shown to accumulate in the spleens of tumor-bearing mice and exert immunosuppressive effects on CD8+ T cells by inhibiting their proliferation, thus diminishing immune responses to pathogens and tumor antigens. In cancer patients with anemia, circulating erythroid cells exhibit similar inhibitory effects, which can be reversed by ROS inhibitors such as apocynin. The adoptive transfer of CD45+ erythroblasts into tumor-bearing mice accelerated tumor growth, whereas the in vivo depletion of erythroblasts using anti-CD71 antibodies suppressed tumor progression and restored immune responses to control levels [6].

Some researchers suggest that CD45+ erythroblasts exhibit stronger immunosuppressive potential due to increased ROS production [6,7]. Furthermore, CD45+ erythroblasts include distinct subgroups based on VISTA expression. VISTA+ CD45+ erythroblasts produce significantly higher ROS levels than VISTA- counterparts, implicating VISTA in immune exhaustion and tolerance pathways [7]. The V-domain Ig suppressor of T cell activation (VISTA) is a novel negative regulator of T cell functions [8].

Erythroblasts are known to produce ROS, arginase, cytokines, and chemokines [9,10,11,12], including TGFb [13]. Transcriptomic changes in erythroblasts under different physiological and pathological conditions have been recently characterized [14,15,16]. However, significant differences exist in the expression of immunomodulatory molecules, including PDL1 and others, among erythroid cells isolated from different sources [7,17]. Also, the role of B-cell surface PDL1 has been described in HIV and non-Hodgkin lymphomas [18]. This suggests that the variations in erythroblast properties may result from stimulation by certain factors, presumably cytokines or components of the microenvironment, which may enhance or diminish the immunosuppressive properties of erythroid cells.

In this study, we focus on the effects of soluble erythroblast-derived factors on immune cells and investigate how these factors modulate lymphoid cell phenotypes.

Aim: To evaluate the impact of soluble products derived from murine erythroblasts on lymphoid cell phenotypes under normal and hematopoietically active conditions and to characterize erythroblast phenotypes based on the expression of immunosuppressive surface and intracellular proteins.

## 2. Materials and Methods

### 2.1. Mice and Experimental Models

Male F1 (CBA × C57Bl6) mice aged 3–5 months were used. All animals were maintained under standard vivarium conditions at the Research Institute of Fundamental and Clinical Immunology (RIFCI). Euthanasia was performed by cervical dislocation after isoflurane anesthesia. Each experimental group included at least six mice (*n* = 6–8). Experimental models (hemolytic anemia, acute blood loss, hypoxia) and erythroblast isolation protocols were previously described [14,15,16]. All procedures were approved by the RIFCI Ethics Committee (protocol No. 129, 17 February 2021). In brief, hemolytic anemia was modeled by the administration of phenylhydrazine (PHZ) (P26252-100G, Sigma-Aldrich, St. Louis, MO, USA) in phosphate-buffered saline (PBS) intraperitoneally: the first injection was 1.2 mg of PHZ per mouse, the second injection was 0.6 mg of PHZ per mouse (after 24 h), and the third injection was 0.6 mg of PHZ per mouse (after an additional 12 h). We performed organ harvesting after 4 days from the start of the experiment [14]. Hypoxia was simulated by placing the mice in a hyperbaric chamber where a negative pressure of approximately ~−46 kPa was maintained for 16 h. Such a pressure corresponds to a rise to an altitude of over 4500 m above sea level. Thus, our model corresponds to high-altitude hypoxia. After the exposure was over, the mice were transferred to the standard conditions of a conventional vivarium. Organ retrieval was performed 3 days after the start of the experiment [15]. Blood loss was performed as follows: In mice under isoflurane anesthesia (Aerran, Baxter, Deerfield, IL, USA), ~0.5–0.8 mL of blood was collected from the retro-orbital sinus using a glass capillary with a pointed tip. The onset of surgical anesthesia and recovery from anesthesia for each animal were monitored. Organs were collected on the third day after the start of the experiment [16]. Intact normal mice without any manipulations were used as controls.

### 2.2. Erythroblast Cells Isolation

We harvested femurs and spleens from mice aseptically. We obtained marrow cells by marrow canal PBS washing via a syringe. Marrow cells were suspended by intense pipetting. Splenocytes were obtained by homogenizing the whole organ in a glass homogenizer. We centrifuged splenocytes in density gradient Ficoll–Urografin (ρ = 1.119 g/cm^3^) for 30 min at 322 RCF and washed them twice in PBS to aim to delete RBC, reticulocytes, and granulocytes. Thus, the obtained fraction of splenic mononuclear cells was used for the next manipulations. Erythroblasts were separated from splenic mononuclear cells via magnetic separation using anti-Ter-119-biotinylated antibodies (#116203, Biolegend, San Diego, CA, USA) and streptavidin-linked magnetic beads (#480015, Biolegend, San Diego, CA, USA), according to the manufacturer’s protocols (MojoSort™ Streptavidin Nanobeads Column Protocol—Positive Selection, Biolegend, San Diego, CA, USA). We measured Ter-119-selected erythroblast viability on a Countess 3 Automated Cell Counter (Thermo Fisher Scientific, Waltham, MA, USA) according to the manufacturer’s protocols using Trypan Blue. Trypan Blue staining showed >94% viability for the sorted erythroid cells.

### 2.3. Cell Culturing and Harvesting the Conditioned Media of Erythroid Cells

We cultured the magnetically sorted Ter-119-positive erythroid cells in the X-VIVO 10 serum-free medium (Lonza, Basel, Switzerland) with the addition of Insulin–Transferrin for 24 h at a seeding density of 1 million per mL of the medium. We collected the conditioned media of the erythroid cells from the cells after 24 h of culturing. We performed the separation by centrifugation at 1500 rpm for 10 min; the cells’ conditioned media were then transferred into new 1.5 mL tubes and frozen at −80 °C. For the next test, we decided to pool Er-conditioned media from all mice in one group (normal, hemolytic anemia, blood loss, hypoxia) to avoid any fluctuations.

### 2.4. Phenotypic Analysis of Lymphoid Cells Exposed to Erythroblast-Derived Soluble Factors

Splenic mononuclear cells from normal intact mice were isolated and cultured in 48-well plates (500,000 cells/well in 500 μL of RPMI-1640 (BioLot, Saint-Petersburg, Russia), supplemented with HEPES, glutamine, gentamicin, benzylpenicillin (all supplements by BioLot, Saint-Petersburg, Russia), and 10% FBS (LtBiotech, Vilnius, Lithuania)). Conditioned media (CM) from erythroblasts (50% final volume) were added to the experimental wells. Control wells received RPMI-1640 without CM.

For proliferation analysis, cells were pre-stained with CFSE. Following 24 h incubation, cells were collected, washed in PBS with 0.09% NaN_3_, and stained with monoclonal antibodies: PacificBlue-CD3 (cat#100214), AlexaFluor488-CD4 (cat#100423), APC/Cy7-CD8 (cat#100714), BV510-CD279 (cat#135241), PerCP-CD19 (cat#115531), and APC-CD30 (cat#102312, all Ab from BioLegend, San Diego, CA, USA). Antibody incubation lasted 20–30 min in the dark at room temperature, after which the cells were washed with 0.5 mL of PBS with 0.09% NaN_3_. Intracellular staining for FoxP3 was performed using fixation/permeabilization buffers. Next, the cells were fixed in 0.5 mL per sample of Fix Buffer (cat#420801, BioLegend, San Diego, CA, USA) containing 4% formaldehyde for 20 min. After fixation and washing out the fixative, the cells were permeabilized in 1 mL per sample of permeabilization buffer, prepared by diluting 10× Perm Buffer (cat#421002 BioLegend, San Diego, CA, USA) 10-fold with distilled water. Permeabilization was performed as follows: 1 mL of 1× permeabilization buffer was added to the cell pellet, vigorously resuspended, and vortexed for ~10 s. The cells were then pelleted by centrifugation at 1500 rpm for 10 min, and the supernatant was removed and replaced with another 1 mL of 1× permeabilization buffer. The cell pellet was vigorously resuspended, vortexed for ~10 s, and centrifuged again at 1500 rpm for 10 min, after which the supernatant was discarded. After permeabilization, AlexaFluor700-FoxP3 (cat#126422 BioLegend, San Diego, CA, USA) monoclonal antibodies were added to the cell pellet at concentrations according to the manufacturer’s instructions. Antibody incubation lasted 20–30 min in the dark at room temperature, after which the cells were washed twice with 1× permeabilization buffer (1 mL per sample). Flow cytometry was conducted using an Attune NxT Flow Cytometer (ThermoFisher Scientific, Waltham, MA, USA). Gating strategy: CD3+ T cells (subgated to CD4+, CD8+, FoxP3+), CD19+ B cells (CD30+ activated and PD1+ suppressed subsets). See more about the gating strategy in Appendix A.

### 2.5. Phenotypic Characterization of Erythroblasts

Cells from bone marrow and spleen (1 × 10^6^ cells/mL) were stained for FITC-TER119 (cat#116206), PacificBlue-CD45 (cat#103126), APC/Cy7-CXCR4 (cat#146524), APC-Galectin-3 (cat#125420), PerCP-Cy5.5-Galectin-9 (cat#136112), PE-Cy7-PDL1 (cat#124314), PE-CCL2 (cat#505904), PE-IL10 (cat#505008), PE-TGFβ1 (cat#141404), and PE-Cy7-IL12/IL23p40 (cat#505210, all Ab from BioLegend, San Diego, CA, USA)). Intracellular staining included fixation/permeabilization (the procedure described above). Flow cytometric analysis identified TER119+ cells, further divided into CD45+ and CD45- subsets for marker profiling. See more about the gating strategy in Appendix B.

### 2.6. Data Analysis

Statistical analysis was performed using GraphPad Prism 8. The Kruskal–Wallis test was used for flow cytometry data. Results are presented as median with interquartile range (Q1–Q3). Differences were considered statistically significant at *p* < 0.05.

## 3. Results

### 3.1. Effect of Erythroblast-Conditioned Media on Lymphocyte Phenotypes In Vitro

Following 24 h incubation with erythroblast-conditioned media (CM), significant phenotypic changes were observed in the lymphoid populations. There was an increased proportion of CD19+PD-1+ B cells in cultures treated with CM from both control and experimental groups, including erythroblasts derived from mice subjected to hemolytic anemia, blood loss, or hypoxia, compared to untreated controls (Figure 1).

Additionally, the proportion of CD3+ T cells among splenic mononuclear cells increased after exposure to CM from erythroblasts derived from both control and hematopoiesis-activated animals (Figure 2). Within the CD3+ population, an elevated percentage of FoxP3+ regulatory T cells was observed, and CFSE staining confirmed enhanced proliferation occurring specifically within the CD3+FoxP3+ subset (Figure 3). CFSE is carboxyfluorescein succinimidyl ester, a molecule that can penetrate into the cell and accumulate in the cytoplasm, and we can see the peaks of CFSE via flow cytometry at cell division. In contrast, no significant alterations were detected in the CD4+/CD8+ T cell ratio (Figure 4a) or in the proportion of CD19+ B cells, activated CD19+CD30+ B cells, CD16|32+ NK-cells, and CD14+ monocyte cells (Figure 4b), indicating that erythroblast-derived soluble factors selectively influence regulatory but not effector lymphocyte populations.

### 3.2. Phenotypic Features of CD45+ and CD45- Erythroblasts

Building upon prior studies by Shahbaz et al. [19], which demonstrated the role of CD71+VISTA+ erythroid cells in promoting Treg development via TGFb, and data from Perik-Zavodskaia et al. [14], revealing the subdivision of murine erythroblasts into CD45+ and CD45- subpopulations, we examined key surface and intracellular mediators to distinguish between these subsets in bone marrow and spleen under normal and hemolytic conditions. In the bone marrow, hemolytic anemia was associated with increased intracellular TGFb expression in both CD45+ and CD45- erythroblasts, with the most pronounced elevation observed in the CD45- subset. Concurrently, IL10 levels decreased specifically in CD45- erythroblasts under these stress conditions, but they were non-significant (Figure 5).

An analysis of surface marker expression under physiological conditions revealed that Galectin-3 was primarily expressed on CD45+ erythroblasts, while CD45- erythroblasts displayed minimal Galectin-3 expression. Both subsets showed a negligible surface expression of Galectin-9 and PDL1 in the absence of stress. Notably, CD45- erythroblasts expressed higher levels of CXCR4 on their surface compared to CD45+ cells. Under hemolytic conditions, CD45+ erythroblasts exhibited a marked reduction in Galectin-3 expression, while both Galectin-9 and PDL1 were significantly upregulated on their surface. Furthermore, a contrasting pattern of CXCR4 expression was observed: levels decreased in CD45- erythroblasts while increasing in CD45+ cells (Figure 6).

In the spleen, erythroblasts demonstrated distinct profiles compared to those in the bone marrow. At normal conditions, splenic CD45- erythroblasts have significantly higher intracellular levels of TGFb compared to their CD45+ counterparts. Under hemolytic conditions, CD45- erythroblasts exhibited significantly lower intracellular levels of TGFb compared to normal CD45- erythroblasts (Figure 7). There are not some significant changes in the intracellular levels of IL10, CCL2 and IL12p40 between all groups.

CD45+ splenic erythroblasts consistently expressed higher levels of PDL1 than CD45- erythroblasts, both under normal and anemic conditions. Hemolytic anemia also triggered a robust upregulation of Galectin-9 expression in CD45+ erythroblasts, further underscoring their shift toward an immunosuppressive phenotype during systemic stress (Figure 8).

## 4. Discussion

The obtained results demonstrate that the increase in CD19+PD-1+ B cells in the presence of erythroblast-conditioned media indicates the potential of erythroid cells to modulate the functional state of B lymphocytes. PD-1 is a marker of functional exhaustion in B cells, suggesting that erythroblasts may suppress B cell activity. The ability of erythroblasts to inhibit B cell proliferation and suppress humoral immune responses in both mice and humans was previously reported [2,3]. In our study, we observed an increase in CD19+PD-1+ B cells in cultures exposed to erythroblast-conditioned media. Elevated PD-1 expression on B cells enhances their sensitivity to its ligand, PDL1. We also noted that CD45+ erythroblasts upregulate surface PDL1 expression during hemolytic anemia, suggesting enhanced immunosuppressive potential. The expression of PDL1 on erythroblasts has been reported in other immune-altered states, including virus-associated solid tumors in humans [7] and in pregnant mice [17].

Additionally, soluble factors secreted by erythroblasts significantly influence T cell differentiation in vitro. One key finding was the increase in CD3+ T cells among splenic mononuclear cells incubated with erythroblast-conditioned media, indicating the enhanced proliferation or survival of this population. Notably, there was also an increase in the proportion of FoxP3+ regulatory T cells (Tregs) among CD3+ cells. CFSE staining confirmed that proliferation occurred primarily within the CD3+FoxP3+ subset, highlighting the selective expansion of Tregs in response to erythroblast-derived factors. This effect may reflect the immunosuppressive role of erythroblasts aimed at maintaining immune homeostasis and preventing excessive immune activation. This concept was previously proposed by Elahi et al. in the context of neonatal immune regulation during the transition from the sterile intrauterine environment to microbial colonization [20]. Moreover, TGFb is known as a potent inducer of Treg differentiation [21], and earlier studies have demonstrated that erythroblasts can produce TGFb [13]. It should be mentioned that some authors suppose that TGFb may mediate in part the suppressive activity of erythroid cells, so authors suggested an involvement of more than one mediator in erythroblast-derived immunosuppression [22].

Our data show a moderate (~10%) but statistically significant increase in CD3+ T cells in erythroblast-CM-treated cultures, likely driven by the emergence and proliferation of CD3+FoxP3+ Tregs. This finding further supports the role of erythroblasts in modulating lymphocyte phenotypes. The ability of erythroblasts to induce Treg differentiation was previously demonstrated in co-culture experiments where splenic erythroblasts from neonatal mice promoted FoxP3 expression in naive CD4+ T cells [19].

Based on the data presented in this study, CD45- bone marrow erythroblasts, particularly under conditions of activated hematopoiesis, appear to be the primary source of TGFb. In contrast, CD45+ erythroblasts increase the surface expression of Galectin-9 and PDL1 during hemolytic anemia, reflecting a shift toward an immunosuppressive phenotype and their potential to modulate T cell responses. Galectin-9 binds to TIM-3 on immune cells, leading to reduced effector functions of cytotoxic T cells, the inhibition of Th1 responses, and the induction of peripheral tolerance [23,24,25].

Although splenic erythroblasts display distinct intracellular immune mediator profiles, CD45+ erythroblasts from the spleen—both in steady state and during hemolytic anemia—exhibit a higher surface expression of PDL1. Notably, Galectin-9 expression increases significantly in CD45+ erythroblasts during hemolytic anemia.

Interestingly, we did not observe significant changes in the CD4+/CD8+ T cell ratio, suggesting that erythroblast-derived factors exert non-selective effects on these subsets. Likewise, the absence of changes in the proportion of activated CD19+CD30+ B cells indicates that erythroblasts do not promote B cell activation under these conditions.

Finally, we observed a redistribution of CXCR4 expression: it was reduced on CD45- erythroblasts and increased on CD45+ erythroblasts during hemolytic anemia. This shift may reflect alterations in migratory properties and the localization of erythroid cells in response to stress stimuli.

## 5. Conclusions

These findings highlight the dual regulatory role of erythroblasts in modulating T and B cell function via both soluble and surface-bound factors. The described mechanisms may contribute to immune homeostasis or suppression under physiological and pathological conditions, such as anemia, inflammation, and malignancy.

## Figures and Tables

**Figure 1 cimb-47-00550-f001:**
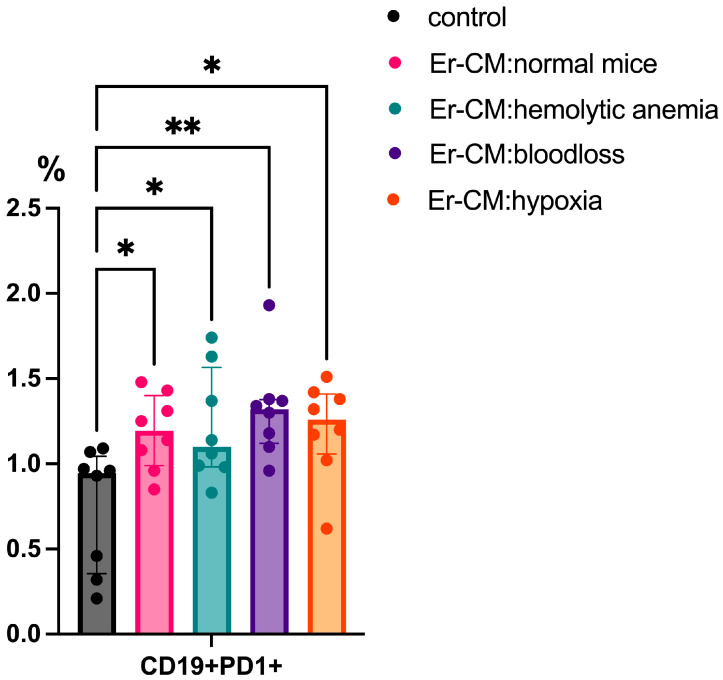
The percentage content of CD19+PD1+ cells in the spleen mononuclear cells after 24 h incubation with erythroblast-conditioned media (Er-CM) and control—RPMI1640. The differences are statistically significant between groups: * *p* < 0.05 and ** *p* < 0.01.

**Figure 2 cimb-47-00550-f002:**
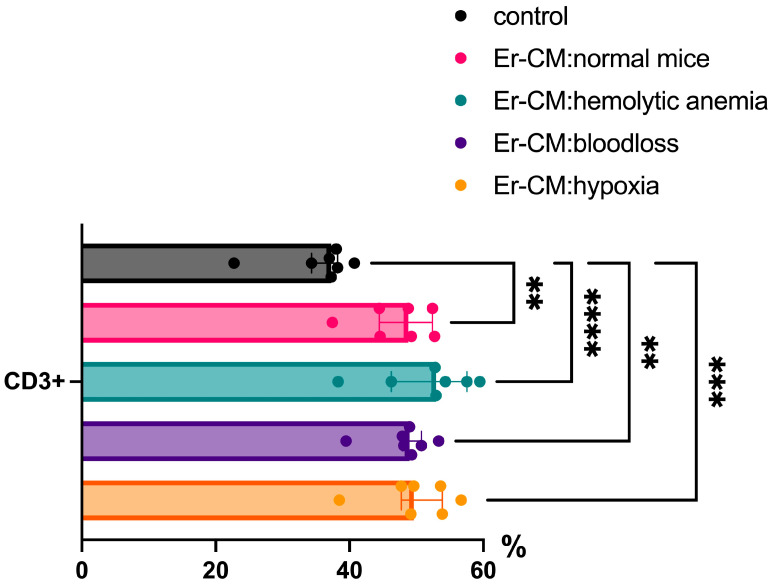
The percentage content of CD3+ cells in the spleen mononuclear cells after 24 h incubation with erythroblast-conditioned media (Er-CM) and control—RPMI1640. The differences are statistically significant between groups: ** *p* < 0.01, *** *p* < 0.005, and **** *p* < 0.001.

**Figure 3 cimb-47-00550-f003:**
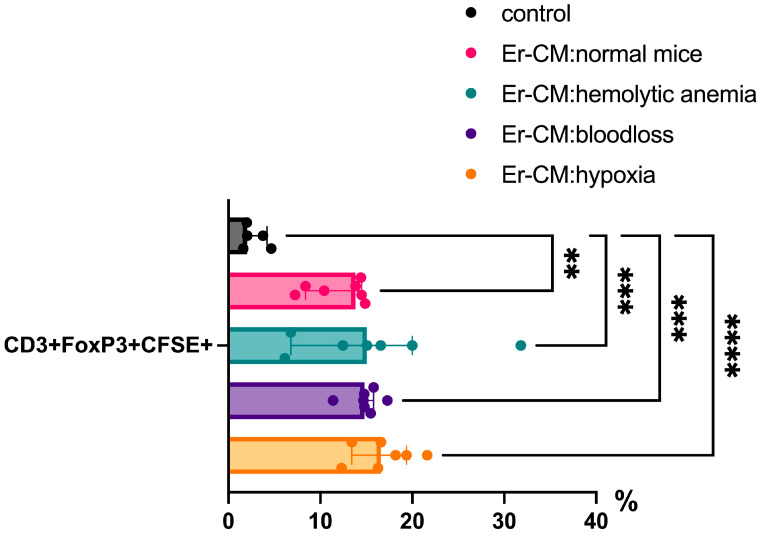
The percentage content of pre-stained CFSE CD3+Foxp3+ cells in the spleen mononuclear cells after 24 h incubation with erythroblast-conditioned media (Er-CM) and control—RPMI1640; The differences are statistically significant between groups: ** *p* < 0.01, *** *p* < 0.005, and **** *p* < 0.001.

**Figure 4 cimb-47-00550-f004:**
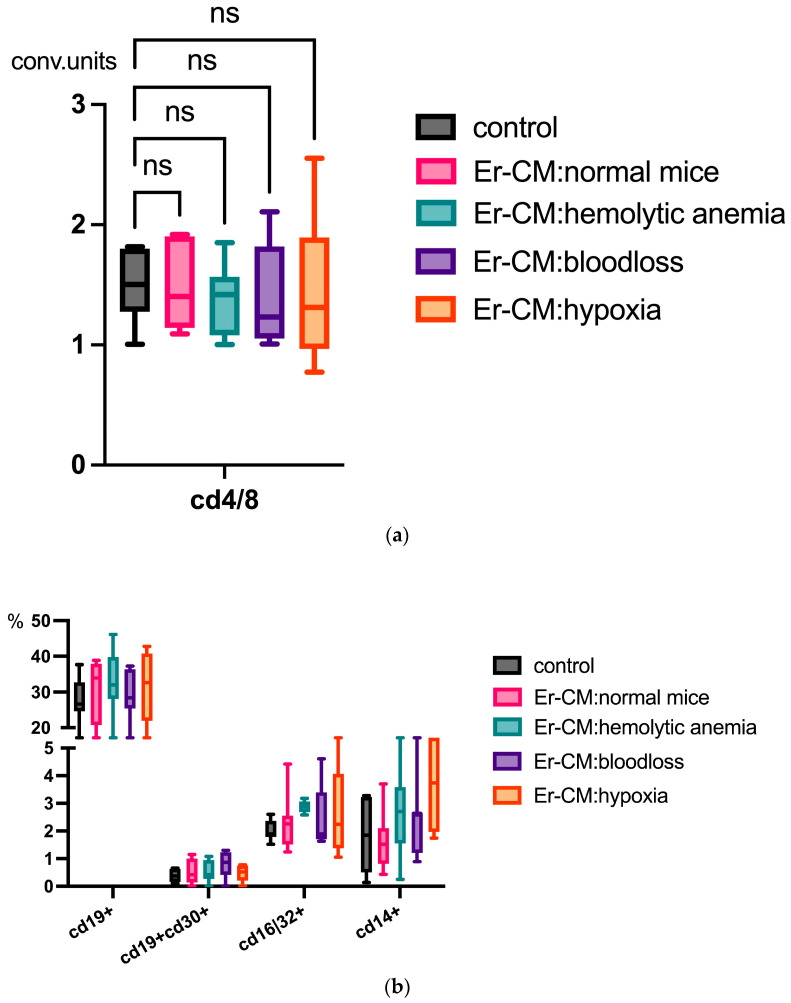
(**a**) No differences in the CD4 cells/CD8 cells ratio in the spleen mononuclear cells after 24 h incubation with erythroblast-conditioned media (Er-CM) and control—RPMI1640; ns = non-significant. (**b**) No differences in the percentage content of CD19+ B cells, activated CD19+CD30+ B cells, CD16|32+ NK-cells, and CD14+ monocyte cells in the spleen mononuclear cells after 24 h incubation with erythroblast-conditioned media (Er-CM) and control—RPMI1640.

**Figure 5 cimb-47-00550-f005:**
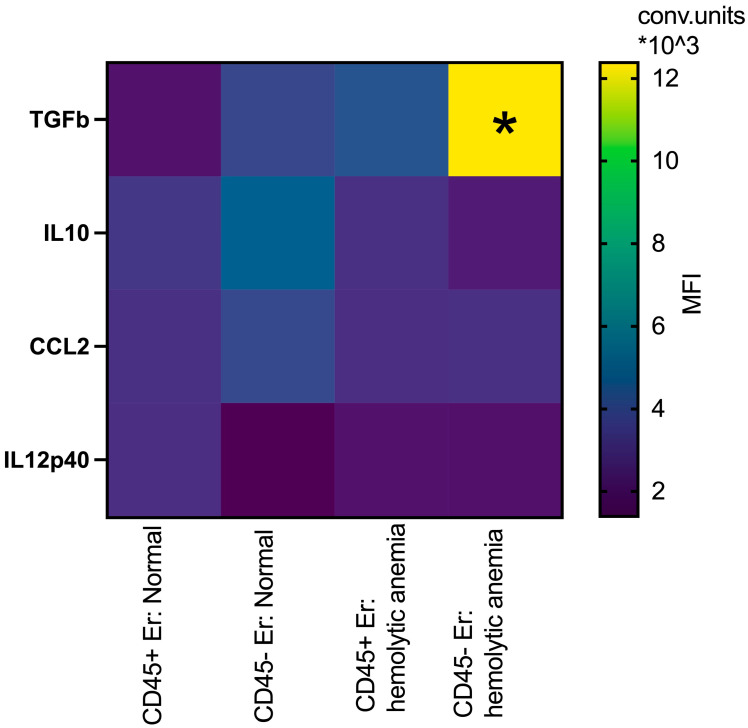
Heatmap: the median fluorescence intensity of intracellular content mediators in CD45+ and CD45- bone marrow erythroid cells at normal conditions and after hemolytic anemia; the differences are statistically significant between groups: * *p* < 0.05 vs. CD45-Er: normal and vs. CD45+Er: hemolytic anemia; conventional units of MFI.

**Figure 6 cimb-47-00550-f006:**
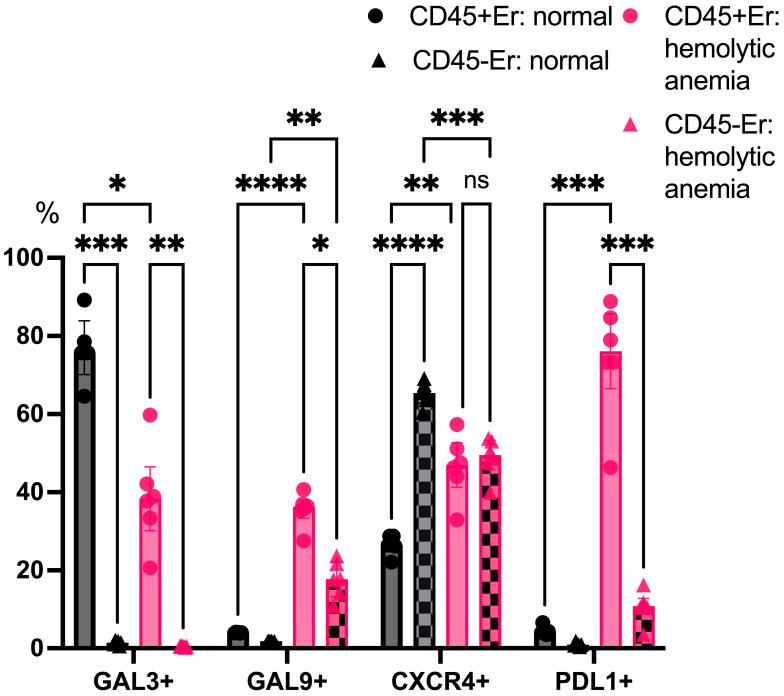
The percentage content of immunoregulatory molecules on marrow erythroid cell surface at normal conditions and after hemolytic anemia; the differences are statistically significant between groups: * *p* < 0.05, ** *p* < 0.01, *** *p* < 0.005, **** *p* < 0.001, and ns = non-significant.

**Figure 7 cimb-47-00550-f007:**
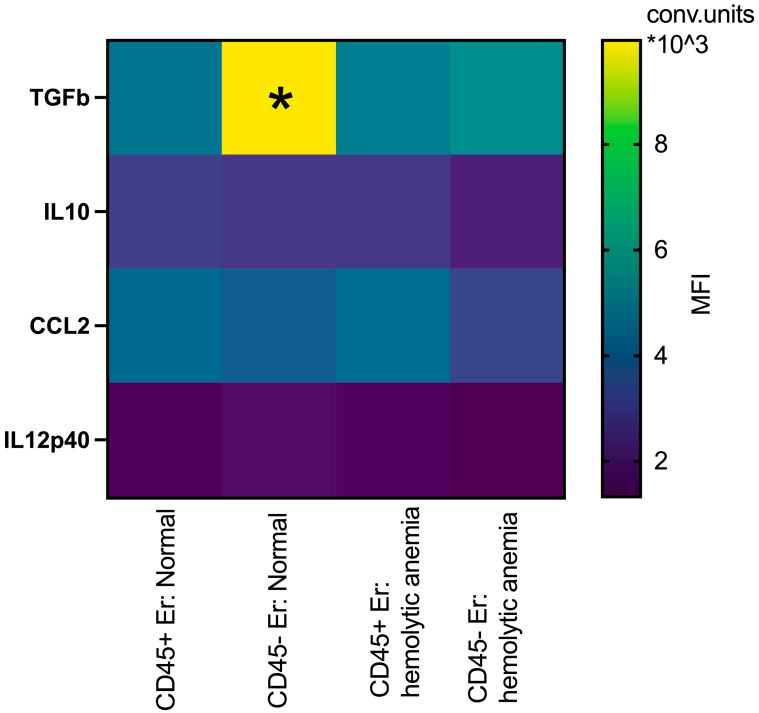
Heatmap: the median fluorescence intensity of intracellular content mediators in CD45+ and CD45- spleen erythroid cells at normal conditions and after hemolytic anemia; the differences are statistically significant between groups: * *p* < 0.05 vs. CD45+Er: normal and vs. CD45-Er: hemolytic anemia; conventional units of MFI.

**Figure 8 cimb-47-00550-f008:**
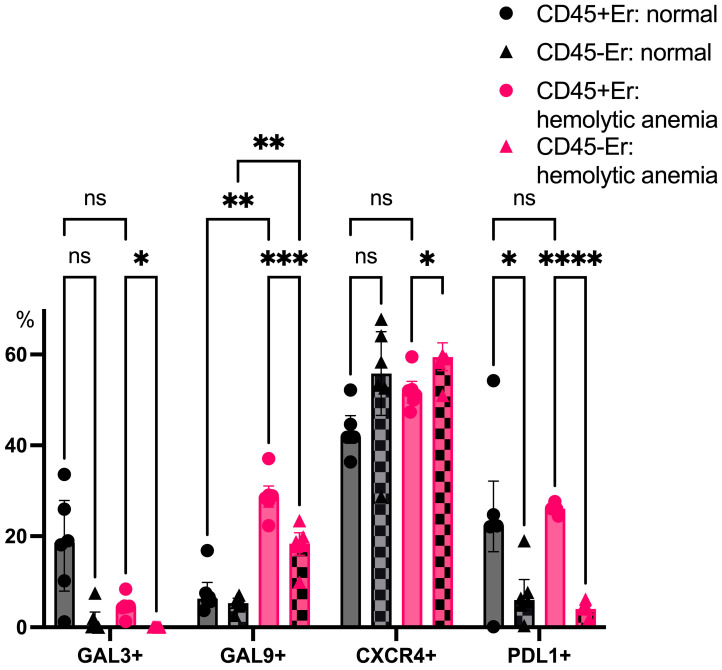
The percentage content of immunoregulatory molecules on spleen erythroid cell surface at normal conditions and after hemolytic anemia; the differences are statistically significant between groups: * *p* < 0.05, ** *p* < 0.01, *** *p* < 0.005, **** *p* < 0.001, and ns = non-significant.

## Data Availability

The data that support the findings of this study are available from the corresponding author, Sergey Sennikov, upon an email request.

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
