# Peer review of "Erythroblasts Promote the Development of a Suppressive Lymphocyte Phenotype via Treg Induction and PD1 Upregulation on the Surfaces of B-Cells: A Study on the Subpopulation-Specific Features of Erythroblasts"

_cimb, 2025, doi:10.3390/cimb47070550_

Round 1
Reviewer 1 Report
Comments and Suggestions for Authors
Manuscript is well prepared apart from some graphical aspects in figures (see below) and a very succinct writing. Conclusions are in general supported by the experiments and somehow new, although of interest for a reduced audience.
Major points
- Authors are strongly recommended to show in a single figure with the same graphical setting all the population they can measure instead of separately showing only significatively affected populations in separate giant figures. Avoid data not shown.
- The same for relevant ratios, put them in a single figure
- Please justify and discuss better the choice of factors examined and the reason not to carry an multi-cytokine assay on conditioned media
- The same for surface markers.
Minor points
ROW 11 untreated mice more than healthy mice
ROW 47 Who is VISTA?
Why are erythroblast cytokines represented in an heatmap and surface markers in a bar graph (with no Y axis label)?
Based on the data presented in this study, CD45⁻ bone marrow erythroblasts, particularly under conditions of activated hematopoiesis, appear to be the primary source of TGFβ. I do think other TGFb producing cells exist in the organism.
Author Response
Dear reviewer
Thank you for your comments and notes. We wanna elucidate some moments about your notes.
Erythroblasts surface markers are presented as the percentage of cells expressing the investigated marker (note that all graphs are labeled as percentages, % and other units - conventional units - where CD4/8 ratio, for example). Intracellular markers are displayed as a heatmap, as this improves visualization—here, we measured fluorescence intensity, and the data are presented as median (MFI).
CD19+PD1+ cells, CD3+ cells, and others are shown on separate graphs due to their distinct quantitative changes. This approach avoids overcrowding the bar plots and improves graph readability.
For example, the plot for CD3+Foxp3+CFSE+ cells shows the percentage of proliferating cells among all CD3+Foxp3+ cells, which is why it is presented in a separate figure. Additionally, in accordance with your request, Figure 4B has been added, displaying previously unreported data.
Of course, I agree with you that there are other TGFβ-producing cells in the body besides erythroblasts. Our primary focus, however, is on studying the properties of hematopoietic and immune cells. As is well known, among immune cells, the main producers of TGFβ are regulatory T cells (Tregs). However, this is a relatively small cell population—in mouse bone marrow, T cells are generally scarce, and Tregs are predominantly found in the thymus, peripheral immune organs (spleen, lymph nodes), and bloodstream.
In contrast, erythroblasts are a highly abundant population in the bone marrow, accounting for 20–45% of all bone marrow cells according to various estimates. And since CD45- erythroblasts represent the major subset among them, we believe that their contribution to TGFβ production in the bone marrow is highly significant.
I wanna say a few words about the analysis of cytokine levels in erythroblast-conditioned media:
In our previous studies, this was performed using BioPlex ImmunoAssay (murine and human erythroblast-conditioned media), and the results are available at the following links: doi:10.3390/cells12242810 doi.org/10.3390/ijms242115752 doi:10.1371/journal.pone.0287793 . Unfortunately, this panel does not allow for TGFβ quantification, as measuring this cytokine is particularly challenging and requires a dedicated reagent kit. We plan to conduct this specific test in future studies. For the current investigation, we opted to analyze intracellular TGFβ levels in erythroblasts using flow cytometry, which represents the most optimal and accessible method available to us at this time.
Reviewer 2 Report
Comments and Suggestions for Authors
Nazarov et al describe an immunoregulatory activity in conditioned media from cultured murine erythroblasts from healthy mice and mice subjected to hypoxia, blood loss and hemolytic anemia. Splenic mononuclear cells cultured with conditioned media shows modulation of T cell differentiation, Treg development and B-cell PD-1 surface expression. Hemolytic stress increases CD45+ erythroblast PDL1 and Galectin-9 and CD45- erythroblast TGFbeta production.
Critique:
Line 47: Define/explain VISTA expression.
Line 57: Define/explain PD-L1 which is distinct from PDL1.
Line 173: Define CFSE.
Line 226 is an incomplete sentence.
What is the sex of mice used in these experiments?
How many mice were used to harvest erythroblasts for each condition (healthy, hypoxia, blood loss and hemolytic anemia)? Were erythroblasts cultured separately for each mouse? How many erythroblast cultures were set up for each condition and was conditioned media pooled or assessed for individual lymphoid cultures? Please clarify.
What is the effect of heat inactivation on the immunoregulatory activity of conditioned media from the erythroblast cultures of healthy mice or from mice subjected to hypoxia, blood loss and hemolytic anemia?
In figure 1, the percentage content of CD19+PD1+ cells is low, as expected, but the variations are also very small – (from 1% to 1.3%) for the various conditioned medial (CM) conditions. Please comment/discuss the potential biological impact of these small changes. These differences are considerably smaller than the variation observed in the elevated numbers of PD-L1 expressing B cells observed in patients – for example, patients with HIV and non-Hodgkin lymphoma (Epeldegui M et al., Scientific reports (2019) 9:9371).
Author Response
Dear reviewer
Thank you for your comments and notes about our manuscript. We wanna elucidate some moments.
Lines 49-50 We added a little bit of information about VISTA with reference.
PDL1 = PD-L1, some authors write this with line (PD-L1) and some authors write this without line (PDL1). We clear this inaccuracy and made this uniform at all our manuscript.
About CFSE lines 174-176 we added some information
Lines 233-234 the sentence was corrected
About sex of mice: all mice were male (please see section 2.1, line 68). This choice was permitted to avoid potential influences of oestrous cycle if we will use female mice.
How many mice were used to harvest erythroblasts for each condition (healthy, hypoxia, blood loss and hemolytic anemia)? Answer: n=6 in each group (normal, hemolytic anemia, bloodloss, hypoxia)
Were erythroblasts cultured separately for each mouse? Answer: yes, we cultured erythroblasts from each mouse separately.
How many erythroblast cultures were set up for each condition and was conditioned media pooled or assessed for individual lymphoid cultures? Answer: for test with splenic lymphoid cells we pooled Er-conditional media from all mice in one group (normal, hemolytic anemia, blood loss, hypoxia) to avoid any fluctuations between mice and get uniform effects. See lines 111-112
About heat inactivation. Under standard culture conditions (37°C, 24 hours), significant heat inactivation of immunoregulatory proteins in the conditioned media is unlikely for several reasons. First of all. Most immune-modulating proteins (TGFb, other cytokines) maintain stability at 37°C for ≤72 hours. Denaturation typically requires 42-56°C or more. Second. Sea of lab protocols (aspecially in vitro induction or differentiation of cells to some needed phenotype) include a few days of culturing with cytokines or any specific protein factors (approx~3-4 days and then it need to replace the media or adding a new dose of factors). Third. We are sure that the cells are recepting cytokines or any factors via receptors, so cytokine or any factor was took fast by cells and then cytokine call the effects. So we suggest there are not some heat inactivation effects of proteins in our Er-conditional media.
We agree with your observation that the increase in CD19+PD1+ cells among splenic lymphoid cells following culture with erythroblast-conditioned media is relatively small. However, we would like to emphasize that these small increments are statistically significant. To better understand this phenomenon, extended culture durations (48, 72, or 96 hours) could be investigated.
Regarding comparative contexts: In HIV infection and non-Hodgkin's lymphoma, studies report more pronounced increases in circulating CD19+PD1+ cells - though it's important to note these represent chronic processes developing over years. As highlighted in a January 2020 correction to one such study, the percentage of CD19+PD1+ cells in patient blood: increases during 1-4 years of disease progression and decreases when disease duration exceeds 4 years doi:10.1038/s41598-019-45479-3
For our experimental model, we propose an alternative approach that might better capture erythroblast immunomodulatory effects:
Pre-immunization: Prime mice with sheep erythrocytes to induce an active immune response (including humoral/B-cell activation; takes 1-2 weeks)
Subsequent analysis: Examine whether erythroblast-derived soluble factors can suppress these activated B-cells in culture